# A Retrospective Study of the Influence of Life Events and Social Support on Relapses and Recurrences in Older Patients with Bipolar Disorder

**DOI:** 10.3390/geriatrics10010016

**Published:** 2025-01-17

**Authors:** Hanna Cusell, Rob Kok

**Affiliations:** Parnassia Psychiatric Institute, Mangostraat 1, 2552 KS The Hague, The Netherlands; hanna.cusell@gmail.com

**Keywords:** older adults, bipolar disorder, life events, social support, relapse, recurrence

## Abstract

**Background/Objectives**: Life events and lack of social support are risk factors for a relapse or recurrence in patients with a bipolar disorder, yet studies focusing on older adults remain limited. Our aim was to investigate the influence of life events and social support on the course of bipolar disorder in older adults. **Methods**: A retrospective cohort study included patients aged 55 years and older in treatment for bipolar disorder and who used lithium. During a follow-up of maximum 5 years, relapses and recurrences, life events and six social support variables were constructed based on patients’ electronic medical files. **Results**: We included 100 older patients with a mean age of 68.1 (SD 8.6) years; 69% were female. At least one relapse or recurrence was observed within the 5 years of observation in 52% of our patients. Life events were noted in the medical files in 24 out of these 52 (46.2%) patients. Living alone, a lower quality of social support and having at least two children was significantly associated with the onset of a relapse or recurrence (*p* = 0.024, *p* < 0.001, *p* = 0.022, respectively). **Conclusions**: Older adults with bipolar disorder have a high rate of relapses or recurrences within 5 years of observation, and half of the relapses or recurrences were preceded by life events. Social factors may also play a significant role in the onset of relapses and recurrences. Our results underline the necessity for incorporating social and environmental factors into prevention of relapses or recurrences for older bipolar patients.

## 1. Introduction

Bipolar disorder is a severe and chronic disorder, with high rates of relapses and recurrences, reduced psychosocial functioning and a loss of approximately 10–20 potential years of life [1]. The recurrence of depressive and (hypo)manic episodes differs greatly among people with bipolar disorder. Although biological and genetic factors play a very important role in the etiology of bipolar disorder, reviews also found an important role for life events and social support [2,3,4,5]. Patients with bipolar disorder reported more life events before relapse when compared to euthymic phases [4]. However, there may be a bidirectional relation between life events and relapses or recurrences, as mania may precede the occurrence of positive life events and depression may precede the occurrence of negative life events [6]. Some studies have found a polarity-specificity effect of life events: negative life events were solely related to depressive episodes, whereas goal-attainment life events were solely related to (hypo)manic episodes [6,7,8,9].

Although less studied, several reviews of social support in bipolar disorder found that social support could act as a protective factor for bipolar patients [2,10]. In addition, mood symptoms may affect social support, also suggesting a bidirectional relation [10]. A polarity-specificity effect of social support has also been suggested in some studies. One study found that social support was not related to the reoccurrence of manic symptoms but was negatively related to the recurrence of depressive symptoms [7]. Another study also found that family functioning and social support did not predict mania in the first year following acute-phase treatment, and that only social support predicted depression [11]. A recent review found that temperament traits such as harm avoidance or novelty-seeking influenced emotional processing and social interactions, which may also influence relapse rates in patients with bipolar disorder [12].

There is a paucity of studies on bipolar disorder in older adults, although this is an even more complex subgroup with prevalent cognitive deficits, increased risk of dementia, frequent physical comorbidities, premature death and high caregiver burden [13,14]. A study of 29 older (mean age 60.1 years) and 56 adult (mean age 34.8 years) patients with a bipolar disorder found that both age groups perceived their social support as inadequate compared with controls of similar age [15]. The same authors found in a later study of 58 older adults with a bipolar disorder (mean age 59.3 years) that negative stressful life events were significantly much more prevalent in patients compared with age-matched controls [16]. In a naturalistic cohort study of 101 older bipolar patients (mean age 68.9 years), no association was found between life events and recurrence during a 3-year follow-up [17]. Finally, a study of 64 older patients (mean age 67.4 years) with a bipolar disorder found that patients who experienced at least one relapse during the follow-up period were less likely to report having children (*p* = 0.04) in univariate analyses [18]. However, in logistic-regression analyses correcting for many confounders, no associations were found between social, psychological and cognitive factors and relapses in bipolar disorder in older patients.

This study aimed to investigate the influence of life events and social support on relapses and recurrences in older patients with bipolar disorder.

## 2. Materials and Methods

### 2.1. Participants

The participants in this retrospective study were in- and outpatients in treatment at Parnassia Psychiatric Institute in The Hague. This study is part of an ongoing research project aimed at the relation between lithium serum levels and relapse or recurrence rate. Patients were included if aged ≥55 years, had a diagnosis of bipolar I or II disorder according to the DSM-IV-TR (APA, 2000) criteria and used lithium. Pharmacy records of all patients prescribed lithium between 2010 and 2015 were assessed as the first step of this study. Patients were excluded if they were hospitalized for (almost) the entire observation period, were observed for 3 months or less, in case of a very unstable course with too many relapses/recurrences in in the case that full remission was not achieved. We excluded these patients because data for these patients did not fit into the primary aim of our study. The study was conducted according to the guidelines of the Declaration of Helsinki and approved by the Institutional Review Board of Parnassia Psychiatric Institute. Since the data were collected from the electronic patient files without any burden for the patients, no additional approval from a research ethics committee or informed consent were needed according to national laws on regulations on medical research.

### 2.2. Procedure

The life-chart method was used to provide a schematic graphical representation of the course of a disease [19]. Life-charts included all bipolar relapses or recurrences between 2010 and 2015, their type/polarity (depressed, mixed, (hypo)manic episode) and duration, and whether life events preceded these episodes, as well as the time between life events and these episodes. Because life events were not often registered in the medical files when no relapse or recurrence occurred, only patients who experienced relapses/recurrences were included in these analyses. We searched the electronic patient files for clues on life events and social support from up to a year before the start of the first relapse/recurrence. For these analyses, also patients who did not experience any relapses/recurrences were included. Six variables related to social support were formed based on medical files. Four of these variables were based on more objective data (living together and number of children, which are both part of the standard intake procedure, and presence of a DSM-IV axis IV code for “problems with primary support group” and for “problems related to the social environment”. The remaining two social support variables were constructed based on notes in the medical files on quality of social support and on loneliness. For example, quality of social support was assessed by searching in the medical files for keywords like partner, husband, child, daughter, brother, and friends, and then for notes about the quality of the social support, according to the patient. Two assessors (both authors) independently rated these last two social variables. The value of these variables was set if the assessors agreed. If there was a disagreement, both assessors studied the medical files again and discussed their ratings until final agreement was achieved.

### 2.3. Materials

The Life Events List by Paykel et al. [20] is a validated method of assessing life events and was used by the first author to score life events mentioned in the medical files. Life events were categorized as being positive (11), neutral (11) or negative (39) according to Koenders et al. [6]. Ten additional life events were found in our study population that were not listed by Koenders et al. We added these after a consensus meeting by four geriatric psychiatrists. Holidays were considered positive life events (*n* = 3), and problems with neighbors, families of children, deterioration of health of a partner, sexual abuse of a child, a sister leaving for a long period, and care for a partner were considered negative life events.

### 2.4. Statistical Analyses

To summarize the demographic statistics of the population, descriptive statistics were used. We used chi-square tests or Fisher exact tests and calculated Odds Ratios (OR) and their 95% Confidence Intervals (CI) for the association between relapses/recurrences and the social support variables.

For the moderator analysis, we used model 1 of the program Process v4.0 by Andrew F. Hayes in SPSS. We investigated whether the interaction effects between the variable ‘life event’ and the different social support variables were significant. If the interaction effect was significant, the direction of the moderation was further explored. We used IBM SPSS 27 and results were considered statistically significant at *p* < 0.05.

## 3. Results

Life charts were available for 129 older patients. For 29 patients, it was not possible to assess social support variables due to a lack of information in the electronic patient files concerning these topics. This resulted in a study group of 100 patients, of which 52 had a relapse or recurrence noted during our observation period and the remaining 48 patients had no notes of a relapse of recurrence. In 27 patients, the first relapse or recurrence during the observation time was a depressive episode, and in 25 patients this first episode was (hypo)manic. The number and type of mood episodes before our observation period was often difficult to assess, because files were digitalized just before our observation period started and the paper files were often not available anymore. No patient was reported to have a mixed episode during follow-up. Table 1 shows the descriptive statistics of our study group.

### 3.1. Life Events and Risk of Relapse or Recurrence

Life events were noted in the medical files of 24 out of the 52 (46.2%) patients who had a relapse or recurrence. In 17 patients, the relapse or recurrence was within 1 month after the life event, in 4 patients it was within 1–3 months, and in 3 other patients it was between 3 and 6 months after the life event. We could not compare this with the occurrence of life events in patients with no relapse or recurrence, because in these patients, medical files seldom had notes about life events as mentioned in the methods section. Due to the low occurrence of positive (*n* = 3) and neutral life events (*n* = 2), we only analyzed whether the association differed between negative life events and a depressive episode (10 negative life events) or a (hypo)manic episode (9 negative life events). This difference was not statistically significant (Chi-2 = 0.9382, df = 1, *p* = 0.9415).

### 3.2. Social Support and Risk Relapse or Recurrence

We analyzed whether our six social support variables were associated with having a relapse or recurrence. For this analysis, we first dichotomized both variables that had three categories and compared low/moderate quality of social support versus good quality of social support and compared having children versus not having children. In 29 patients, we could not find any information about having children in their medical files. As can be seen in Table 2, the results were significant for two out of our six social support variables. The results were also statistically significant for the variable quality of life if all three categories (good, moderate and low) were used, Chi-2 = 13.838, df = 2, *p* < 0.001), but were not significant if three categories for having children (0, 1 or ≥2) were used (Chi-2 = 5.126, df = 2, *p* = 0.077). However, a post hoc analyses of having 0–1 child versus ≥2 children showed a significant association with having a relapse or recurrence (Chi-2 = 5.124, df = 1, *p* = 0.022) with an OR for relapse/recurrence for patients having ≥2 children versus 0–1 child of 3.5 (95% CI 1.2–10.4). Feeling lonely was not significantly associated with having children or not (Chi-2 0.065, df = 1, *p* = 0.799). Two-third of the patients living together felt lonely, versus 44% of the patients living alone (Chi-2 = 5.537, df = 1, *p* = 0033).

We calculated the OR for having a first depressive episode versus a first (hypo)manic episode during our observation period for the 6 social support variables. The results are presented in Table 3. None of our social support variables were significantly associated with a higher chance of a depressive episode versus a (hypo)manic episode.

### 3.3. Social Support as a Moderator Between Life Events and Relapse or Recurrence in Bipolar Disorder

To determine whether one or more social support variables influenced the relation between the independent variable ‘life event’ and the dependent variable ‘type of relapse’, six moderator analyses were carried out and are presented in Table 4. There were no significant interaction effects, indicating that none of the social support variables influenced the relation between life events and type of relapse.

## 4. Discussion

The aim of this study was to investigate the influence of life events and social support on new depressive, (hypo)manic or mixed episodes in euthymic older patients with a bipolar disorder. In our study group, 52% experienced at least one relapse or recurrence during the follow-up of maximum 5 years of observation, almost equally divided in depressive and (hypo)manic episodes. This is in line with the 39.1% of older bipolar patients who reported at least one recurrence during the 3-year follow-up period [18]. We found that in almost half of our patients with a relapse or recurrence, a life event was noted in the medical files in the year before a relapse or recurrence occurred. In the vast majority, the life event occurred within a month before the relapse or recurrence. Due to the retrospective nature of our study, we could only analyze whether life events preceded new depressive or (hypo)manic episodes, but not what the actual role was in eliciting a new episode, because life events were rarely noted when no relapse or recurrence occurred. We were therefore not able to replicate the results of the only other study of life events and recurrences in older patients with a bipolar disorder, which found no association between life events and recurrence during a 3-year follow-up [17]. In that study, most patients (68.8%) reported having experienced positive or negative life events in the 3 years of follow-up. This number may be higher compared to our patients, because all patients were interviewed, and this differs from our study, as we only checked our patients’ medical files for information on life events. Underreporting of life events in medical files may play a role, as we expect that some patients may not have reported life events to their treating physicians or nurses, or if they did, these life events may not have been reported in the patient’s files. Unfortunately, the only other study of stressful life events in older bipolar patients that we are aware of did not report the number of life events and its association with a relapse or recurrence [16]. In line with our results is that even in interviews, positive life events were hardly mentioned [17]. In our study, negative life events preceded both depressive and (hypo)manic episodes. Due to the lack of positive life events, we could not analyze whether there was a polarity-specific association between life events and relapses or recurrences in older patients, as has been found in adult patients [6,7,8,9].

Living alone, low quality of social support and having ≥2 children were significantly associated with a relapse or recurrence. Other studies also found that older adults have decreased perceptions of social support compared to older controls [15]. One other study in older bipolar patients found, in univariate analyses, an significant association between having no children and recurrence However, after correcting for confounders, no significant association was found between social factors (including having a partner, children, and social support) and having a recurrence during the follow-up period [18]. The negative association with having at least two children and having more relapses or recurrences in our study may appear at first sight to be counterintuitive. Our post hoc result may be a spurious finding and needs replication before any conclusion can be drawn. However, in the patient files of our studies, it became apparent that the quality of the relationships between patients and their children was sometimes almost non-existent or even quite negative. This may be the result of children having to cope with, or even protecting themselves from, disruptive interactions with their parents when they are suffering from a depressive or manic episode. Patients with bipolar disorder may have high expressed emotions, increased family dysfunction and interpersonal conflicts [15]. Levels of familial expressed emotion may predict depressive symptoms [5]. Psychological variables may help explain the vulnerability of bipolar patients to social environments, including personality factors (e.g., neuroticism), reward sensitivity, and difficulty with the accurate perception of facial emotions [5]. Personality factors such as affective temperament may also influence the course of bipolar disorder [12].

In line with this finding is that significantly more patients living together felt lonely compared to patients living alone. We hypothesize that patients suffering from bipolar disorder may feel lonely if the spouse is not able to support the patient enough during their mood episodes, which may of course be very difficult. Nevertheless, our results indicate the importance of social factors during bipolar disorder in older adults and suggest that this is an important topic for those responsible for the care of older adults with bipolar disorders. Improving these social adversities, with, for example, psychoeducation or relationship counseling, may prevent relapses or recurrences in bipolar disorder [5], has not been studied in older adults, as far as we know.

We could not find a significant effect of our social support variables in the onset of depressive versus (hypo)manic episodes during our observation period. This is in contrast with studies in adults, where social support seems to be linked to fewer relapses in depression but not in (hypo)mania [7,11]. However, the low number of patients in our analyses may explain our negative results.

We expected that social variables would act as a buffer against life events but could not find an interaction effect between these variables. However, this may be a result of lack of power. Personality factors such as, for example, temperament, may also interact with social support and life events, but our retrospective study lacks data about personality factors.

### Strengths and Limitations

The present study has various strengths and limitations. The data of all older bipolar patients in treatment in our mental health institute who met the inclusion criteria stated in the methods section were analyzed. Therefore, our study results may be more generalizable to the real-world population compared to randomized controlled trials or prospective longitudinal studies, both of which suffer from a large selection bias. This is also the first study that investigated the association between life events, social support and their interaction with relapses or recurrences in older patients with a bipolar disorder, to our knowledge.

Some limitations may be taken into account. First, life events were only found in the patient’s files when a relapse or recurrence occurred and may be underreported in these patients, because this variable was based on notes in medical records. In addition, a causal relation between a life event and relapse or recurrence is difficult to establish. Second, due to the retrospective nature of our study, we were not able to use validated questionnaires or interviews to assess life events or social support variables in a more objective manner. Two of our social support variables, quality of social support and loneliness, were estimated by a subjective process based on case notes in the medical files. However, this variable was constructed by the rating of two independent reviewers of the medical files. All other social support variables were more objective, but underreporting may still play a role. If, for example, loneliness was not reported, it remains unclear whether the variable was absent or merely undocumented. However, we searched in the files during an observation period of up to 5 years in which patients typically had very many case notes by psychiatrists, psychologists and nurses, and it is not very likely that these variables were never noted, if present. A third limitation is that patients’ reflections of their life events and support systems may have been influenced by a depressive or (hypo)manic episode. For example, during a depressive episode, a patient may rate social support more negatively than during euthymic periods. Finally, our sample size may have been too small for some analyses and lack of power may explain some of our negative findings, as, for example, in our moderator analyses.

## 5. Conclusions

Older adults with bipolar disorder have a high rate of relapses or recurrences within 5 years of observation, of which half is preceded by life events. However, our study first needs replication in a larger study group with a prospective study design, with validated rating scales to assess social support variables. We suggest that mental health workers should be aware that after a life event, many older adults with a bipolar disorder have a new onset of a relapse or recurrence. During euthymic phases, adverse social support may be a relevant topic to discuss with older bipolar patients and their relatives, as these variables may also be risk factors for a new relapse or recurrence.

## Figures and Tables

**Table 1 geriatrics-10-00016-t001:** Demographic and clinical variables of the study group (*n* = 100).

Variable	Total (*n* = 100)	No rel/rec (*n* = 48)	Depressive rel/rec (*n* = 27)	(Hypo)manic rel/rec (*n* = 25)
Age in years, mean (sd)	68.1 (9.0)	67.2 (8.6)	69.6 (8.3)	68.3 (10.7)
Gender, female, *n* (%)	69 (69.0)	35 (72.9)	19 (70.4)	15 (60)
Life event				
-positive, *n* (%)	3 (3.0)		2 (7.4)	1 (4)
-neutral, *n* (%)	2 (2.0)		1 (3.7)	1 (4)
-negative, *n* (%)	19 (19.0)		10 (37.0)	9 (36)
Living alone, *n* (%)	64 (64.0)	25 (52.1)	21 (77.7)	18 (72)
Quality of social support *				
-good, *n* (%)	37 (37.0)	26 (52.1)	6 (22.2)	5 (20)
-moderate, *n* (%)	47 (47.0)	18 (37.5)	17 (63.0)	12 (48)
-bad, *n* (%)	15 (15.0)	3 (6.3)	4 (14.8)	8 (32)
Loneliness, yes, *n* (%)	15 (15.0)	5 (10.4)	8 (29.6)	2 (8)
Children **				
-none, *n* (%)	24 (33.8)	14 (40)	5 (27.8)	5 (27.8)
-one, *n* (%)	26 (36.6)	15 (42.9)	6 (33.3)	5 (27.8)
-two or more, *n* (%)	21 (29.6)	6 (17.1)	7 (38.9)	8 (44.4)
DSM-IV ‘problems primary support group’, yes, *n* (%)	58 (58.0)	25 (52.1)	19 (70.4)	14 (56)
DSM-IV ‘problems social environment’, yes, *n* (%)	34 (34.0)	15 (31.3)	11 (40.7)	8 (32)

*n* = number, sd = standard deviation, rel = relapse, rec = recurrence, * information was present in 99 files, ** information was present in 71 files.

**Table 2 geriatrics-10-00016-t002:** Associations between social support and relapse/recurrence (*n* = 100).

Social Support Variable, *n* (%)	Relapse/Recurrence		OR (95% CI), *p*-Value
	Yes (*n* = 52)	No (*n* = 48)	
Living alone	39 (71.2)	26 (54.2)	2.5 (1.1–5.9), *p* = 0.024
Low/moderate quality of social support *	41 (78.9)	26 (54.2)	4.4 (1.8–10.6), *p* < 0.001
Loneliness	10 (19.2)	5 (10.4)	2.1 (0.6–6.5), *p* = 0.269
Having children **	26 (72.2)	21 (60)	1.7 (0.6–4.7), *p* = 0.322
DSM-IV ‘problems with primary support group’	33 (63.5)	25 (52.1)	1.6 (0.7–3.6), *p* = 0.312
DSM-IV ‘problems related to social environment’	19 (36.5)	15 (31.3)	1.3 (0.6–2.9), *p* = 0.674

OR = Odds Ratio, CI = Confidence Interval, * information was present in 99 files, ** information was present in 71 files (36 with relapse/recurrence, 35 without).

**Table 3 geriatrics-10-00016-t003:** Associations between social support and type of episode (*n* = 52).

Social Support Variable	OR for Depressive Versus (Hypo)manic Episode
Living alone	1.4 (95% CI 0.4–3.5), *p* = 0.631
Low/moderate quality of social support	0.9 (95% CI 0.2–3.3), *p* = 0.845
Loneliness	4.9 (95% CI 0.9–25.6), *p* = 0.078
Having children *	1.0 (95% CI 0.2–4.3), *p* = 1.0
DSM-IV ‘problems with primary support group’	1.9 (95% CI 0.6–5.9), *p* = 0.282
DSM-IV ‘problems related to social environment’	1.5 (95% CI 0.5–4.6), *p* = 0.513

df = degrees of freedom, OR = Odds Ratio, CI = Confidence Interval, * information was present in 36 medical files (18 depressive episodes, 18 (hypo)manic episodes).

**Table 4 geriatrics-10-00016-t004:** Interaction-effects of social support variables and life events.

Moderator Variable	b Interaction-Effect	z	df	*p*
Living alone	−0.3719	−0.2796	3	0.7797
Quality of social support:				
-average versus good	14.6267	0.0209	5	0.9833
-low versus good	16.9109	0.0241	5	0.9807
Loneliness	−14.8870	−0.3620	3	0.9867
Children:				
-one child versus no children	−0.6931	−0.3620	5	0.7174
-two or more children vs. no children	−1.2040	−0.6730	5	0.5009
DSM-IV ‘problems with primary support group’	−0.8597	−0.5935	3	0.5528
DSM-IV ‘problems related to social environment’	0.6549	0.5502	3	0.5822

df = degrees of freedom.

## Data Availability

For ethical reasons, the privacy-sensitive data that support the findings of this study are not publicly available.

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
