# Peer review of "A Retrospective Study of the Influence of Life Events and Social Support on Relapses and Recurrences in Older Patients with Bipolar Disorder"

_geriatrics, 2025, doi:10.3390/geriatrics10010016_

Round 1

Reviewer 1 Report

Comments and Suggestions for Authors

The article investigates the influence of life events and social support on relapses and recurrences in older adults with bipolar disorder. Through a retrospective study of 100 patients aged 55 and older, the authors found that life events and specific social factors, such as living alone and poor quality of social support, significantly increased the likelihood of relapses or recurrences. The study emphasizes the role of psychosocial factors in managing bipolar disorder in older populations and suggests the importance of addressing social adversity in treatment strategies.

Please find several suggestions for modifications

Abstract: Add a concluding sentence emphasizing the importance of addressing social support in clinical management. For example:"These findings underline the necessity of incorporating social and environmental factors into relapse prevention strategies for older bipolar patients."

Introduction:

  • Please incorporate the reference by  Favaretto et al. on the role of temperament traits:

Please find only an example: "As highlighted in the recent review by Favaretto et al. (2023), temperament traits such as harm avoidance or novelty seeking influence emotional processing and social interactions, potentially shaping the risk of relapse in bipolar disorder."

Favaretto E, et al.Synthesising 30 years of clinical experience and scientific insight on affective temperaments in psychiatric disorders: State of the art. J Affect Disord. 2024 Oct 1;362:406-415. doi: 10.1016/j.jad.2024.07.011.

Methods: Please clarify the rationale for excluding patients with a very unstable course;

      Please, include details on how inter-rater disagreements were resolved in assessing social support variables.

Discussion:

Truy to deeper discuss the potential influence of negative parent-child relationships:

It could be valuable to add a table or graph illustrating the timing of life events relative to relapses to provide visual clarity.

The discussion effectively interprets results but could delve deeper into the interaction between temperament, social support, and life events.

Limitations:

The limitations are well-addressed but could benefit from more specific acknowledgment of potential biases in medical record reporting. Add a sentence that could recognize this issue such as (only for example): "The reliance on medical records may lead to underreporting of life events, particularly in cases without a relapse or recurrence." Please, acknowledge the small sample size's impact on moderator analysis results.

Grammar and Stylistic suggestions

Line 18:Please rephrase "but research in older adults is scarce" with "yet studies focusing on older adults remain limited."

Line 68: Please change "less likely to have children" to "less likely to report having children."

Line 230: Please replace "contra-intuitive" with "counterintuitive."

Line 243: Please change "protecting themselves from" to "shielding themselves against."

Line 271: Please modify "we are not sure whether it was not present or not noted" to "it remains unclear whether the variable was absent or merely undocumented."

Author Response

Thank you very much for helping us to improve our manuscript

Abstract: Add a concluding sentence emphasizing the importance of addressing social support in clinical management. For example:"These findings underline the necessity of incorporating social and environmental factors into relapse prevention strategies for older bipolar patients."

We have added: Our result underline the necessity of incorporating social and environmental factors into prevention of relapses or recurrences for older bipolar patients. 

Introduction:

  • Please incorporate the reference by  Favaretto et al. on the role of temperament traits:

Please find only an example: "As highlighted in the recent review by Favaretto et al. (2023), temperament traits such as harm avoidance or novelty seeking influence emotional processing and social interactions, potentially shaping the risk of relapse in bipolar disorder."

We have added: A recent review found that temperament traits such as harm avoidance or novelty seeking influenced emotional processing and social interactions, which may also influence relapse rate in patients with bipolar disorder [12].

Favaretto E, et al. Synthesising 30 years of clinical experience and scientific insight on affective temperaments in psychiatric disorders: State of the art. J Affect Disord. 2024 Oct 1;362:406-415. doi: 10.1016/j.jad.2024.07.011.

Methods: Please clarify the rationale for excluding patients with a very unstable course.

We have first added: (our study was) aimed at the relation between lithium serum levels and relapse or recurrence rate. And added also: We excluded these patients because data of these patients did not fit into the primary aim of our study. 

Please, include details on how inter-rater disagreements were resolved in assessing social support variables.

We have added: (If there was a disagreement), both assessors studied the medical files again (and discussed their ratings until final agreement was achieved).  

Discussion:

Try to deeper discuss the potential influence of negative parent-child relationships.

We have added: Patients with bipolar disorder may have high expressed emotions, increased family dysfunction and interpersonal conflicts [15]. Levels of familial expressed emotion may predict depressive symptoms [5]. Psychological variables may help explain the vulnerability of bipolar patients to social environments, including personality factors (e.g., neuroticism), reward sensitivity, and difficulty with the accurate perception of facial emotions [5]. Personality factors such as affective temperament may also influence the course of bipolar disorder [12].  In line with this finding is that significantly more patients living together felt lonely compared to patients living alone. We hypothesize that patients suffering from bipolar disorder may feel lonely if the spouse is not able to support the patient enough during their mood episodes, which may of course be very difficult. 

It could be valuable to add a table or graph illustrating the timing of life events relative to relapses to provide visual clarity.

We categorised the time between the life events and relapse or recurrence in within one month, between 1-3 months and between 3 and 6 month. This is because medical notes rarely were very precise, and more often this time is expressed as "2 or maybe 3 months ago", or "last month". As the vast majority was within a month, we believe a graph would not be very helpful.  

The discussion effectively interprets results but could delve deeper into the interaction between temperament, social support, and life events.

Unfortunately we do not have data about temperament or other personality factors. We have added: We expected that social variables would act as a buffer against life-events but could not find an interaction effect between these variables. However, this may be a result of lack of power. Personality factors as for example temperament, may also interact with social support and life events, but our retrospective study lacks data about personality factors.

Limitations:

The limitations are well-addressed but could benefit from more specific acknowledgment of potential biases in medical record reporting. Add a sentence that could recognize this issue such as (only for example): "The reliance on medical records may lead to underreporting of life events, particularly in cases without a relapse or recurrence." Please, acknowledge the small sample size's impact on moderator analysis results.

We have added: (life events)  may be underreported in these patients, because this variable was based on notes in medical records.  And also added: (our ... lack of power may explain some of our negative findings), as for example in our moderator analyses.  

Grammar and Stylistic suggestions. These have all been changed.

Line 18: Please rephrase "but research in older adults is scarce" with "yet studies focusing on older adults remain limited."

Line 68: Please change "less likely to have children" to "less likely to report having children."

Line 230: Please replace "contra-intuitive" with "counterintuitive."

Line 243: Please change "protecting themselves from" to "shielding themselves against."

Line 271: Please modify "we are not sure whether it was not present or not noted" to "it remains unclear whether the variable was absent or merely undocumented."

Reviewer 2 Report

Comments and Suggestions for Authors

The paper by Cusell et al reports that in older individuals with bipolar disorder relapses are related to low quality of social support having more than 1 child, and living alone.  My enthusiasm about this paper was slightly reduced due to the reasons listed below:

1. A better justification is needed for why eligible participants had to be on Lithium.

2. The language could be improved. Phrases like "All patients using lithium between 2010 and 2015 were retrieved from the pharmacist of this institute" are confusing. Maybe it is worth pointing out that the pharmacy records were accessed as a part of this study.

3. It is important to indicate whether the hospital has questions about family members and social support as a part of the intake documentation. Otherwise, it is unclear how the information was collected.

4. The authors indicate that "This resulted in a study group of 100 patients, of which 27 patients had a first depressive episode and 25 patients a first (hypo)manic episode during our observation period." Please clarify what the 'first episode' means in the context of this paper. As far as I understand, all participants had been diagnosed with bipolar disorders before the hospital admission.

5. It would be great if the authors clarified whether there is an overlap between those who felt lonely and those with more than 1 child. In general, I believe that the information about loneliness, social support, and the number of children could be better related. This would help explain why people with more than 1 child had a high probability of relapsing.

Author Response

Thank you very much for helping us to improve our manuscript. 

Comment 1. A better justification is needed for why eligible participants had to be on Lithium.

We have added: (our project was) aimed at the relation between lithium serum levels and relapse or recurrence rate.

Comment 2. The language could be improved. Phrases like "All patients using lithium between 2010 and 2015 were retrieved from the pharmacist of this institute" are confusing. Maybe it is worth pointing out that the pharmacy records were accessed as a part of this study.

We have changed this in: Pharmacy records of all patients prescribed lithium between 2010 and 2015 were assessed as first step of this study. 

Comment 3. It is important to indicate whether the hospital has questions about family members and social support as a part of the intake documentation. Otherwise, it is unclear how the information was collected.

We have added: (living together and number of children) are both part of the standard intake procedure. Also we explained how the 2 other variables were constucted and added: For example, quality of social support was assessed by searching in the medical files for keywords like partner, husband, child, daughter, brother, and friends and then for notes about the quality of the social support, according to the patient.  

Comment 4. The authors indicate that "This resulted in a study group of 100 patients, of which 27 patients had a first depressive episode and 25 patients a first (hypo)manic episode during our observation period." Please clarify what the 'first episode' means in the context of this paper. As far as I understand, all participants had been diagnosed with bipolar disorders before the hospital admission.

Patients were not all admitted to a hospital, and could have been diagnosed with bipolar disorder before our observation period started.  We have rewritten this paragraph as follows:  This resulted in a study group of 100 patients, of which 52 had a relapse or recurrence noted during our observation period and the remaining 48 patients had no notes of a relapse of recurrence. In 27 patients the first relapse or recurrence during the observation time was a depressive episode and in 25 patients this first episode was (hypo)manic. The number and type of mood episodes before our observation period was often difficult to assess because files were digitalized just before our observation period started and the paper files were often not available anymore. 

Comment 5. It would be great if the authors clarified whether there is an overlap between those who felt lonely and those with more than 1 child. In general, I believe that the information about loneliness, social support, and the number of children could be better related. This would help explain why people with more than 1 child had a high probability of relapsing.

This was a great idea and we found therefore another interesting result. We have added in the results: Feeling lonely was not significantly associated with having children or not (Chi-2 0.065, df=1, p=0.799). Two-third of the patients living together felt lonely, versus 44% of the patients living alone (Chi-2= 5.537, df=1, p= 0033).  

And in the discussion we added: In line with this finding is that significantly more patients living together felt lonely compared to patients living alone. We hypothesize that patients suffering from bipolar disorder may feel lonely if the spouse is not able to support the patient enough during their mood episodes, which may of course be very difficult. 

Round 2

Reviewer 1 Report

Comments and Suggestions for Authors

Thank you very much for the modifications provided according to revision.